# Novel Weft-Knitted Strain Sensors for Motion Capture [note 1]

**DOI:** 10.3390/mi15020222

**Published:** 2024-01-31

**Authors:** Susanne Fischer, Bahareh Abtahi, Mareen Warncke, Carola Böhmer, Hans Winger, Carmen Sachse, Johannes Mersch, Eric Häntzsche, Andreas Nocke, Chokri Cherif

**Affiliations:** 1Institute for Textile Machinery and High Performance Material Technology (ITM), Faculty for Mechanical Science and Engineering, Technische Universität Dresden, 01062 Dresden, Germany; susanne.fischer2@tu-dresden.de (S.F.);; 2CeTI—Cluster of Excellence, Centre for Tactile Internet with Human-in-the-Loop, Technische Universität Dresden, 01062 Dresden, Germany

**Keywords:** weft-knitted strain sensor, textile-based strain sensor, functional electrical stimulation (FES), multiple sclerosis (MS), knee angle, functional leggings

## Abstract

Functional electrical stimulation (FES) aims to improve the gait pattern in cases of weak foot dorsiflexion (foot lifter weakness) and, therefore, increase the liveability of people suffering from chronic diseases of the central nervous system, e.g., multiple sclerosis. One important component of FES is the detection of the knee angle in order to enable the situational triggering of dorsiflexion in the right gait phase by electrical impulses. This paper presents an alternative approach to sensors for motion capture in the form of weft-knitted strain sensors. The use of textile-based strain sensors instead of conventional strain gauges offers the major advantage of direct integration during the knitting process and therefore a very discreet integration into garments. This in turn contributes to the fact that the FES system can be implemented in the form of functional leggings that are suitable for inconspicuous daily use without disturbing the wearer unnecessarily. Different designs of the weft-knitted strain sensor and the influence on its measurement behavior were investigated. The designs differed in terms of the integration direction of the sensor (wale- or course-wise) and the width of the sensor (number of loops) in a weft-knitted textile structure.

## 1. Introduction

Wearables have developed very rapidly and evolved significantly over the past decades, and, despite some hurdles to market maturity, some products are now integral parts of our daily lives [1,2,3,4,5,6,7]. Due to their convenience and flexibility, they are being intended for use in gesture and posture tracking in healthcare, biomechanical, and physiological monitoring systems [8,9,10,11,12].

An important functional therapy method for multiple sclerosis (MS) or stroke-related weak foot dorsiflexion (foot lifter weakness) is functional electrical stimulation (FES), in which the muscle is activated by the electrical stimulation of the fibular nerve and can, thus, perform its physiological movements [13]. Neuro-prostheses are one of the advances that have been made in the field of FES therapy. They detect the gait phases of the wearer through inertial measurement units (IMUs), which then trigger electrical impulses to stimulate the corresponding nerve at the appropriate moment for dorsiflexion [14,15]. There are commercially available systems on the market [16,17,18,19]. Nevertheless, for daily use, those systems have comfort-limiting factors like their bulky design, the built-in hard electronics, and the limited washability. Therefore, the aim of the current research is the realization of a textile-based sensory and therapeutic stimulative functional system integrated into a weft-knitted fabric in the form of functional leggings.

The gait analysis for FES devices requires a real-time diagnostic function to determine the actual gait phase. While walking, the knee angle changes in accordance with the flexion of the leg. Therefore, the textile structure in the knee area undergoes straining depending on the flexion angle. Integrated textile strain sensors in the knee area are therefore designed to derive the gait phase from the straining of the textile structure. Knitting [20,21], stitching [20,22,23], and embroidering [24,25] are the standard textile methods for producing textile-based strain sensors. In the case of stitched or embroidered sensors, the stretchability of the sensor is largely determined by the basic textile fabric. In contrast, weft-knitted structures have an inherent good stretchability. In addition, weft-knitted strain sensors can be directly integrated into the garment in one single process step during manufacturing; stitched and embroidered sensors require additional process steps to apply the sensor onto the garment.

Weft-knitted strain sensors made of resistive yarns have different parameters that affect the sensor properties and measurement behavior. These can be classified into three categories: the knitting structure, the used yarn material, the sensor geometry, and integration direction. 

The knitting structure included different base structures in weft-knitting. A fundamental distinction must be made between knits produced on one needle bed and knits for which two needle beds are required. Right–left knits, like a single jersey (or plain), are produced on a single needle bed. Right–right knits, like a double jersey (or 1 × 1 rib) or interlock, are produced on two needle beds. The influence of the base structure on the sensor behavior was investigated in [26] by comparing interlock and plain structures manufactured with and without an elastic yarn. Test samples manufactured with an elastic yarn showed that the gauge factor of a resistive yarn-based strain sensor integrated in the plain structure is slightly higher compared to the interlock structure. In the case of linearity, the interlock structure shows better results than the plain structure. Overall, it can be concluded that the use of an elastic yarn has a higher beneficial effect, especially concerning the working range and the gauge factor, than the base structures do on their own.

The used materials included silver-plated polyamide yarns as well as stainless-steel filament or fiber yarns, either bare or blended with non-conductive fibers, which have been widely used as conductive material for weft-knitted strain sensors in different studies. Among them, silver-plated polyamide yarns seem best suited to achieve a good sensor response when tensile-loaded, whereas blended stainless-steel yarns result in sensor signals with more noise and a lower reproducibility [20]. A direct comparison of the sensor behavior between silver-plated polyamide yarns and blended stainless-steel fiber yarns during cyclic tensile testing comes to a similar conclusion. While the sensors with silver-plated polyamide yarns showed better results concerning the change in resistance in response to strain, the sensors made of stainless-steel fiber yarns worsened significantly over time [26] or did not work reliably at all [27]. The additional use of elastic yarns, compared to the sole use of silver-plated polyamide yarns in the sensor area, enlarges the working range of the sensor significantly, as well as increasing its gauge factor [26]. The combination of elastic yarns and silver-plated polyamide yarns also produces very stable sensors suitable for a high number of cycles of cyclical tensile loading [28].

The sensor geometry includes the shape of the sensor and its aspect ratio. In [29], the sensor performance of different polygonal-shaped sensors was compared with rectangular-shaped sensors. The rectangular-shaped sensor showed better sensor performance during a cyclic test regarding higher repeatability and less noise. The numbers of courses and wales determine the aspect ratio of rectangular-shaped sensors. In [30,31], weft-knitted test samples with varying numbers of courses and wales were manufactured. The change in resistance was measured during both static [30] and tensile loading [31]. The tensile loading was in the course direction, and it was shown that the number of courses and wales has an influence on the gauge factor of the sensors. The gauge factor increases with an increasing number of wales and decreases with an increasing number of courses. This would mean that a sensor with a high number of wales and a small number of courses would show the best possible sensor behavior concerning the gauge factor. However, this is contradicted by [29,32]. In [32], samples with a number of courses between one and five were compared. It was found that the number of courses has no significant influence on the hysteresis value. However, regarding the gauge factor, samples with a course number of four showed the highest values. Again, in [29], the “thinnest” and “longest” sensors did not achieve the highest sensitivity regarding the gauge factor. An aspect ratio mostly between 24:1 and 77:1 is recommended. So far, the influence of the integration direction (course-wise or wale-wise integration) on the sensor behavior under tensile loading has not been investigated in depth. From the work of [33], it can be concluded that the straining direction has an influence on the gauge factor. But only square-shaped, not rectangular-shaped, knitted strain sensors were stretched in the course as well as in wale direction. In most studies, the integration direction of rectangular knitted strain sensors is in the course direction and the strain is also applied in the course direction [26,27,28,29,32,34].

Based on the information from the named literature as well as the application of the sensor in functional leggings, some design recommendations for the knitted strain sensor, concerning the knitting structure, the used materials, and the sensor geometry, can already be derived. Table 1 shows these design recommendations as well as their resulting additional benefits for the production of the functional leggings and their wearing comfort.

For the development of the textile-based sensor integrated in the functional leggings, further investigations concerning the integration direction and the width of the sensor are necessary. To date, there is no clear recommendation regarding the direction of integration and exposure to tensile loading corresponding to the integration direction. In addition, there is no clear recommendation concerning the sensor width, especially when the sensor is integrated in the wale-wise direction. Both result in a different number and location of contact points between neighboring sensor loops made of conductive yarn. These can be located at the head, at the legs, and at the feet of the loops (cf. Figure 1).

It is expected that the different number and location of the contact points in the wale and course directions as well as the chosen sensor width result in a different sensor behavior under dynamic strain. The results of the integration direction concerning the sensor performance are especially interesting to determine the subsequent production direction of the leggings, that is, whether it is manufactured vertically (starting with the lower leg and knitting to the waist or vice versa) or horizontally (starting knitting at the side of the leg). The width of the sensor has an influence on the later appearance of the sensor. The narrower the sensor can be designed, the more unobtrusive its appearance in the garment.

## 2. Materials and Methods

The test samples were designed to have a basic weft-knitted fabric, which surrounds the inserted sensor area. The knitting structure is a plain right–left weft-knitted structure (single jersey) for both areas. The basic fabric is made of two combined yarns: a single covered elastic yarn made of LYCRA^®^ 78 dtex and PA66 textured 78f23/1 dtex with a final yarn count 104 dtex (made by Jörg Lederer GmbH, Amstetten, Germany) and a lyocell yarn (TENCEL™ lyocell fiber yarn; yarn count Nm 40). The lyocell yarn acts as the main yarn to build up the knit structure, and it is plated with the single covered elastic yarn. The sensor area was manufactured by using electrically conductive yarn that is plated with the single covered elastic yarn used for the basic knitting fabric. Silver-tech+ 150 by Amann & Söhne GmbH & Co. KG (Bönnigheim, Germany) acts as the sensor. That yarn has a resistance of less than 300 Ω/m and a yarn count of 220 dtex [35]. The binding between the basic knitting fabric and the sensor area was realized by intarsia knitting. The test samples differ in two parameters concerning the sensor area in order to be able to investigate the sensor performance:The integration direction of the sensor: course-wise (horizontal) direction or wale-wise (vertical) direction.The width of the sensor: sensor widths of one, two, or four loops.

The sensor in design A is integrated in the course-wise direction and has a sensor width of one loop or one course, respectively. The sensors in design B, C, and D are integrated in the wale-wise direction and have a sensor width of one, two, and four wales, respectively. Five samples were manufactured for each sensor design. All weft-knitted strain sensors were realized on a conventional weft-knitting machine ADF 530-32 BW knit and wear^®^ (KARL MAYER STOLL Textilmaschinenfabrik GmbH, Reutlingen, Germany) with E14 machine gauge, available at ITM. Table 2 summarizes the materials and bindings of the four different designs.

The sensors are to be integrated above the kneecap of the later leggings, as shown in Figure 2a. In this area, the gait movement results in alternating straining of the textile, which corresponds to the respective knee angle. To best mimic this alternating change in length by the cyclical tensile loading due to the gait movement, the samples are to be cyclically loaded with a tensile testing machine. In order to determine the expected change in length of the weft-knitted structure above the kneecap for the tensile tests, a weft-knitted structure was fixed over the knee joint of a test person. The test person bent the knee in 45 ° steps from the unbent to the fully bent state (cf. Figure 2b).

The maximum angle at full flexion of the knee joint was 130°. The lengths L1 to L4 of the weft-knitted structure for the bent angles 0° to 130° were measured. The measured lengths and the corresponding strains are shown in Table 3. The maximum angle of 130° corresponds to 50% strain of the textile structure in the sensor area. 

For applying the cyclic tensile loads, a tensile tester Junior Z2.5 (ZwickRoell GmbH & Co. KG, Ulm, Germany) was used. The strain rate was set to 1000 mm/min. The tensile strain was increased from 0 to 50% successively in 5% and 10% steps. Every step was repeated for 10 cycles. This was carried out for the loading phase up to 50% strain as well as for the unloading phase back to 0% (cf. Figure 3a). The changes in ohmic resistance of the strain sensors integrated in the test samples were measured simultaneously with the applied tensile load with a precision resistance meter DAQ 6510 (Keithley Instruments Inc., Solon, OH, USA) with the 4-wire method. The clamps for the 4-point measurement were attached to the conductive yarn ends at the opposing ends of the knitted sensor. Figure 3b shows the sample in the tensile test setup. The test program of the cyclic tensile test was deliberately designed to be very extensive. For the results of the present work, the focus was on the measurement results of the ten cycles with 50% elongation, since these cover the maximum necessary tensile load. 

In order to compare the sensor behavior of the different designs, different specific sensor values were determined. They included the unstrained resistance or base resistance R_0_ and the gauge factor (GF). The base resistance R_0_ was determined for the respective test sample at the beginning of the first cycle of 50% strain. For a better comparability, the measured base resistance R_0_ of the full sensor length was calculated for the length per cm (R_0_/cm). The gauge factor (GF) can be calculated by the quotient of the relative electrical resistance change (∆R/R_0_) and the relative change in length (∆l/l_0_) due to the applied tensile strain ε. However, this implies a linear behavior of a resistive strain sensor. Textile sensors usually do not show an ideal linear behavior. To determine the gauge factor, the slope of the regression line of the relative electrical resistance change against the relative change in length was used for this reason.

## 3. Results

### 3.1. Base Resistance R_0_

In Table 4, the mean values and the standard deviation (SD) of the base resistance R_0_ of the different designs are presented. The values for the designs A and B—both have a sensor width of one loop but differ in their integration direction—show that the sensor integrated in the course-wise direction has a resistance of 14.69 Ω/cm, which is significantly higher than the resistance of 3.77 Ω/cm for the sensor integrated in the wale-wise direction. The sensors integrated in the wale-wise direction with sensor widths of two loops (design C) and four loops (design D) have a resistance of 2.51 Ω/cm and 1.61 Ω/cm, respectively.

### 3.2. Relative Resistance Change during Tensile Loading

The plots in Figure 4, Figure 5, Figure 6 and Figure 7 show the relative change in resistance of the test samples under tensile strain. All designs respond to a tensile loading with a change in the simultaneously measured electrical resistance. However, sensors with design A, integrated in the course-wise direction, respond only for strain levels below 10%. Above that threshold, no significant increase in resistance is observed. The samples integrated in the wale-wise direction react over a larger strain range beyond 10% and with a significantly higher change in resistance due to tensile loading. It can also be seen that the relative change in resistance is increasing with the increasing number of wales. On the other hand, the plots show that the linearity of the sensor signal decreases with an increasing number of loops of the sensors in the wale-wise direction, and the hysteresis increases as well. In particular, design D shows two significant hysteresis loops, below and above 30% strain. Below 30% strain, the relative change in resistance increases continuously as the strain increases. Above 30%, the change in resistance starts to decrease for further increasing strain. The hysteresis loop above 30% strain also occurs in a weakened form with design C.

### 3.3. Gauge Factor

The gauge factor (GF) values and the standard deviation (SD) of the different sensor designs are calculated using the software MATLAB R2018b (The MathWorks Inc., Natick, MA, USA). The calculated values are listed in Table 5. Design D has the highest gauge factor value, calculated at 2.48. The gauge factor values for designs A, B, and C have been calculated at 0.04, 1.65, and 2.41, respectively.

## 4. Discussion

The base resistance R_0_ of sensor design A, at 14.69 Ω/cm, is significantly higher than the base resistance of design B, at 3.77 Ω/cm. On the one hand, this can be explained by the different number of contact points between the loops of the conductive sensor yarn. In the course-wise direction, as already described in [34] for interlock-based structures, the possible contact points are at the legs and heads between adjacent loops along the knitted course (cf. Figure 8a).

In the wale-wise direction, the possible contact points are at the heads, legs, and feet of adjacent loops at the interlacing between courses. Every interlacing can be considered as two contact points [36], which doubles the amount of possible contact points (cf. Figure 8b) compared to the course-wise direction. Each contact point increases the number of connections, enabling a sufficient current flow between the loops and not just along the conductive sensor yarn itself. On the other hand, for the same sensor length in the wale-wise direction, more loops are necessary than in the course-wise direction. In the sample design, we aimed for a comparable sensor length, which resulted in 200 loops for the wale-wise direction and only 100 loops for the course-wise direction. This leads to an additional increase in the number of contact points relative to the sensor length for the sensor integrated in the wale-wise direction.

Comparing the sensor designs integrated in the wale-wise direction and different sensor widths with each other, it can be seen that the base resistance is decreasing with an increasing number of loops in the sensor width. As design B, with a sensor width of one, loop has a value of 3.77 Ω/cm, a doubling in the number of loops in the sensor width decreases the base resistance to 2.51 Ω/cm (design C) and 1.61 Ω/cm (design D), respectively. Figure 8c,d show the expected increase in possible contact points due to the increase in loops along with sensor width and the related increase in the amount of conductive sensor yarn. However, doubling the number of loops does not halve the resistance. One reason for this could be that not every contact point between loops guarantees a sufficiently stable current flow. Another reason could be the yarn loops at the sides of the sensor (cf. Figure 8b–d). These side loops are formed when knitting from one course to the next to realize the redirection of the yarn between the courses. Therefore, additional yarn material is needed, which can also enable a current flow and lower the base resistance. For small sensor widths, the amount of additional used yarn for the side loops is higher in relation to the amount of loops for the actual sensor. This could have a reducing effect on the measured resistances, especially for small sensor widths.

Design A, integrated in the course-wise direction, responds only in the lower strain range up to 10% with a change in resistance. That is the reason for the lower GF calculated at 0.04, related to the total strain range of 50%. In comparison, the samples integrated in the wale-wise direction (designs B, C, D) react over a larger strain range beyond 10% and with a significantly higher change in resistance to tensile loading, which seems to be dependent on the number of loops used according to the sensor width. This is also reflected in the calculated values of GF, which are 1.65, 2.41, and 2.48 for designs B, C, and D, respectively. It can be assumed that the increased number of possible contact points between the loops has an influence on both the base resistance and the relative change in resistance. The plots of the strain-dependent relative resistance change also reveal a second hysteresis loop for designs C and D in the range above 30% strain, where the relative resistance change starts to decrease as the strain increases further.

In addition to the influence of different numbers of contact points described above, the influence of elastic yarn must be taken into account to explain these effects. The use of elastic yarn in knitted fabrics results in more compact structures. By plating conductive yarn with elastic yarn, the loops of the conductive yarn are compressed more tightly in the unloaded state. This not only results in a larger amount of conductive yarn in relation to the length of the sensor, but it can also be assumed that this leads to wider and more contact points and thus to better electrical contact between the loops. This effect is illustrated in Figure 9a for the course-wise integration and in Figure 10a for the wale-wise integration of the sensor.

The contact points of the knitted sensors integrated in the course-wise direction separate under tensile load successively (cf. Figure 9b) and the electrical resistance increases. From a certain degree of elongation, there are no more contact points between the yarn (Figure 9c), and a further increase in the resistance due to the contact points is no longer possible. For design A, this degree of elongation seems to be reached at about 10% strain.

Sensors integrated in the wale-wise direction have a significantly higher number of contact points at the heads and the feet due to the integration direction. However, due to plating with elastic yarn, it can be assumed that, in the unstrained state, contact points will mainly occur on the legs of the adjacent loops. Adjacent loops in the wale-wise direction are more highly compressed by the elastic yarn, which creates more contact points in their leg area (cf. Figure 10a). In the semi-loaded state, those contact points are separated successively (cf. Figure 10b) and the measured resistance rises. This continues until the additional conductive yarn material in the loops is consumed by the elongation and the heads and feet of adjacent loops come into closer contact. From this strain level onwards, no additional increase in resistance can be achieved by separating contact points. On the contrary, it can be supposed that a further increase in strain presses the heads and feet of adjacent conductive loops closer together (cf. Figure 10c), resulting in a better electrical contact, which in turn lowers the electrical resistance. The effect of decreasing electrical resistance under applied tensile strain was already reported in [36] for weft-knitted strain sensors in a single jersey consisting of conductive yarn manufactured without elastic yarn. This effect could explain the decreasing resistance values in the range of the second hysteresis loop above 30% strain for designs C and D (cf. Figure 5, Figure 6 and Figure 7). Design B, however, does not show decreasing resistance values in the range above 30% strain, although the occurrence of closer contact between loop heads and feet should also have an effect here. It is possible that the small sensor width of one loop allows for the conductive yarn to slip out of the side loops while strain is applied. This would provide the sensor loops with additional yarn material, which in turn allows for a higher strain range before the loop heads and feet of the sensor come into close contact.

## 5. Conclusions

This work presents a novel approach for the integration of conductive yarns into a weft-knitted structure acting as strain sensor for monitoring and detecting the gait phase of its wearer for FES applications, e.g., for patients suffering from multiple sclerosis. The effect of the integration direction and the number of loops in the sensor width on the electromechanical behavior of the weft-knitted strain sensor during cyclic tensile loading was investigated. It was shown that the different number and location of the contact points of the sensor loops, which result from a different integration direction and number of loops in the sensor width, can give a good explanation for the different results in the observed sensor behavior. The specific sensor values were compared in terms of base resistance, relative change in resistance, and the gauge factor. 

Base resistance:-The base resistance of the sensor integrated in the course-wise direction and with a sensor width of one loop is 14.69 Ω/cm and thus significantly higher than the base resistance of the sensor integrated in the wale-wise direction and with a sensor width of one loop, at 3.77 Ω/cm.-The base resistance for the sensors integrated in the wale-wise direction decreases with an increasing number of loops for the sensor width. The sensor widths of one, two, and four loops result in resistances of 3.77 Ω/cm, 2.51 Ω/cm, and 1.61 Ω/cm, respectively.

Relative change in resistance:-The sensor integrated in the course-wise direction (design A) shows a good correlation between the measured change in resistance and the applied tensile load up to a 10% strain.-The sensor designs integrated in the wale-wise direction have a clear correlation between applied tensile strain and measured resistance change over the majority of the strain range. That covers a range up to 30% strain with the designs of two and four loops (designs C, D) for the sensor width, and a range up to 50% with the design of one loop (design B).

Gauge factor:-The gauge factor of the sensor integrated in the course-wise direction (design A) is very low, at 0.04.-Higher gauge factors can be achieved with the sensors integrated in the wale-wise direction. An increase in the number of loops in the sensor width increases the gauge factor of the sensor. Sensor widths of one, two, and four loops result in gauge factors of 1.65, 2.41, and 2.48 (designs B, C, D), respectively.

Based on these results, the following recommendations can be concluded for the further development of the weft-knitted sensory leggings regarding the sensor design:-The integration direction of the sensor is preferably the wale-wise direction, due to the clear correlation between applied tensile strain and measurable change in resistance over the majority of the strain range as well as the higher gauge factors. This also determines a vertical production direction of the leggings.-A sensor width of one loop (design B) is recommended for a strain range up to 50%. This corresponds to a knee angle of 130° and is the preferred maximum working range of the sensory leggings. In addition to the enlarged strain range, this small sensor width makes the sensor, in terms of its appearance, even more unobtrusive.-Sensor widths of more than one loop increase the gauge factor, enabling a more accurate determination of the knee angle. However, increasing the number of loops also reduces the strain range in which the sensor responds with a clear correlation between measured change in resistance due to the applied tensile strain.

For the further development of the sensory leggings, possible influences on the sensor behavior should also be considered that result from the fact that the leggings are a wearable device. The leggings should not only be able to be worn at constant ambient temperatures, as to be expected of a domestic environment, but walking movements also take place outdoors when the ambient temperature is not constant. In addition, the leggings are worn close to the body; therefore, they are exposed to both the body temperature and possible perspiration, and therefore sweat. Fluctuations in temperature and humidity might affect the sensors’ straining behavior. Because it is a resistive principle, changes in conductivity are of particular importance. On the one hand, moisture, especially sweat due to its salt content, can increase the electrical conductivity of the knitted sensor. On the other hand, it is likely that the effect of moisture on the knitted fabric causes the contact points of the loops to behave differently under applied tensile loads. The sliding friction of the yarns, which is required to change the contact points between the loops, can be affected by this. Furthermore, the leggings should be able to be worn during the entire daily active gait movement phase of their wearer. This results in considerably more load cycles than the number investigated in this study and makes further investigations into long-term cyclic loading necessary. The durability, particularly with regard to possible damage of the sensor due to abrasion, should also be considered in subsequent work. The leggings are designed in order to be worn under normal everyday clothing as well. This causes friction between the layers of fabric during movement, which can affect the sensor properties. Furthermore, the leggings are also an item of clothing, so they should be cleaned regularly. The effect of washing on the sensor properties should also be investigated.

So far, this work has shown promising results of weft-knitted structures acting as strain sensors, which can be used for FES applications. Further optimization of the knitted strain sensor will focus on the improvement of the sensor behavior under long-term stress. The integration in the wale-wise direction and a sensor width of one loop results in a clear correlation between measurable change in resistance up to 50% of applied tensile strain. But these sensors have a lower gauge factor than broader sensor widths. On the other hand, a sensor width of four loops has a comparably high gauge factor but no clear correlation between the measured change in its resistance and the applied strain up to 50%, which reduces the working range of the sensor. One further focus in future work, besides the above-mentioned topics, will be the combination of these properties: a high gauge factor and a wide working range at the same time.

## Figures and Tables

**Figure 1 micromachines-15-00222-f001:**
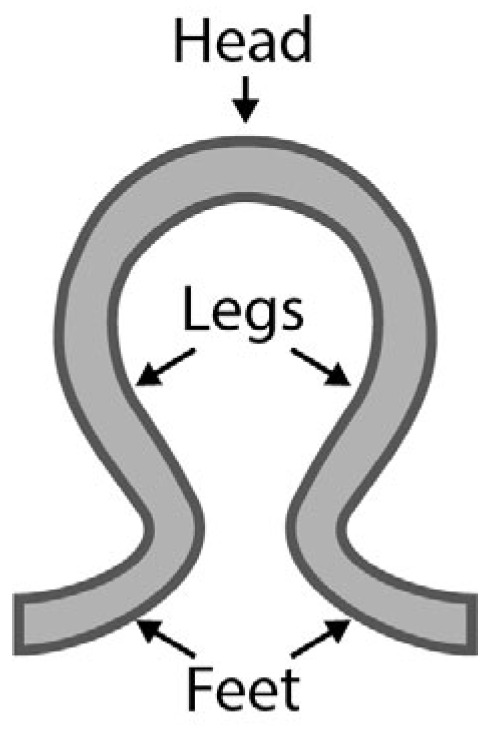
Parts of a knitted loop: head, legs, and feet.

**Figure 2 micromachines-15-00222-f002:**
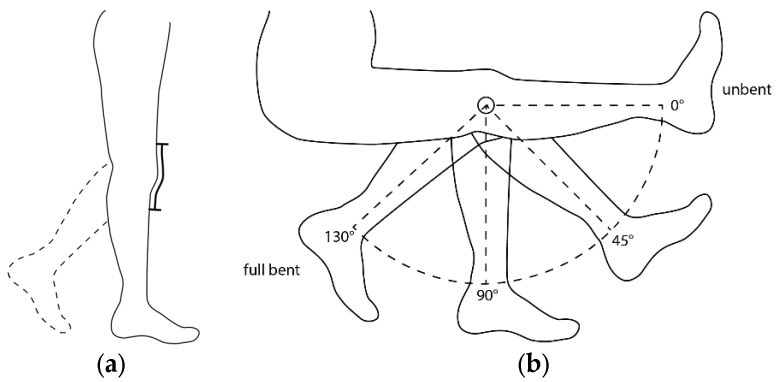
Intended position of the sensor in later leggings (**a**) and leg angulation from unbent to fully bent (**b**).

**Figure 3 micromachines-15-00222-f003:**
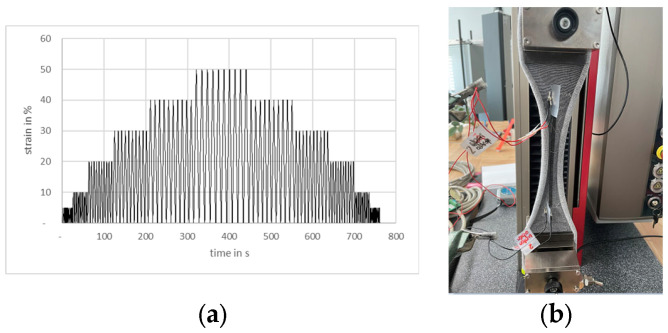
Cyclic tensile test program (**a**) and knitted strain sensor in tensile test setup (**b**).

**Figure 4 micromachines-15-00222-f004:**
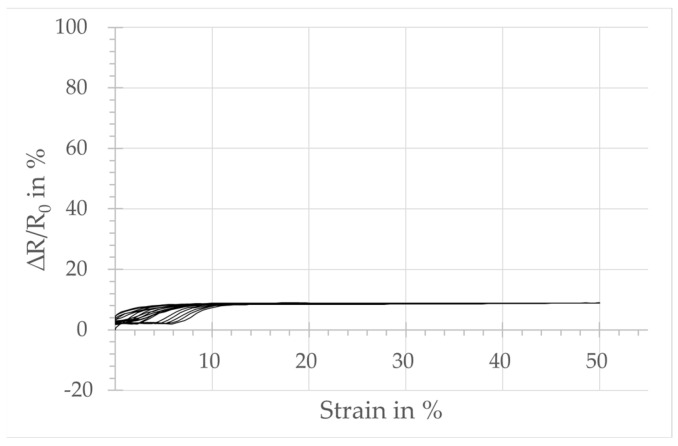
Sensor behavior of strain sensor design A: change in the relative resistance depending on the tensile strain.

**Figure 5 micromachines-15-00222-f005:**
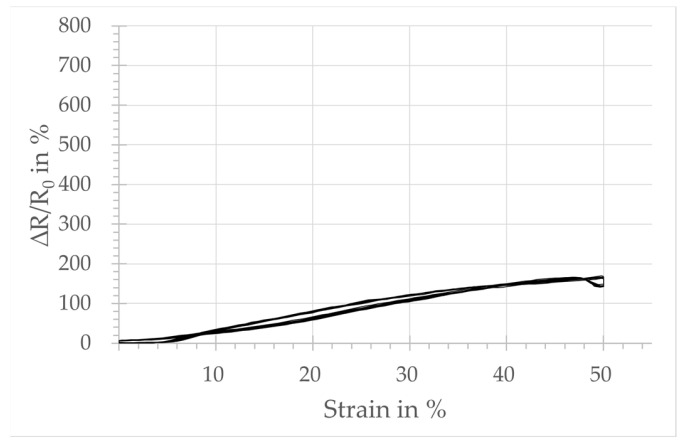
Sensor behavior of strain sensor design B: change in relative resistance depending on the tensile strain.

**Figure 6 micromachines-15-00222-f006:**
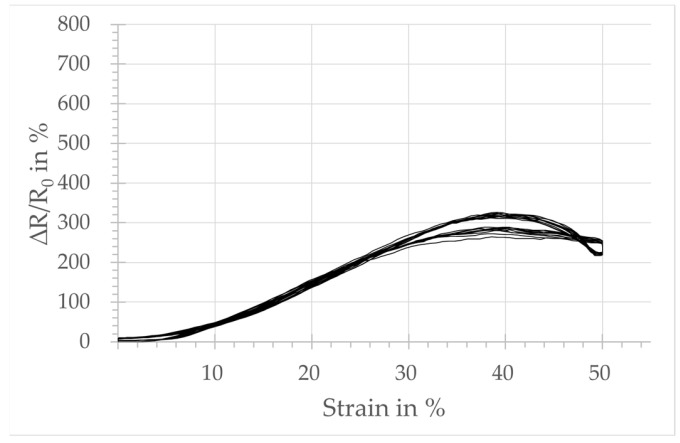
Sensor behavior of strain sensor design C: change in relative resistance depending on the tensile strain.

**Figure 7 micromachines-15-00222-f007:**
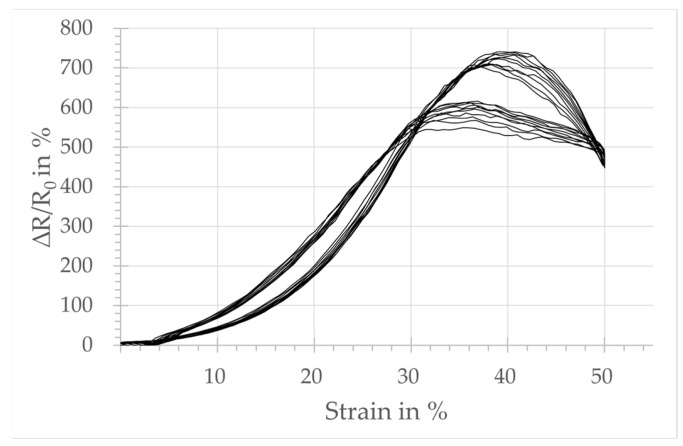
Sensor behavior of strain sensor design D: change in relative resistance depending on the tensile strain.

**Figure 8 micromachines-15-00222-f008:**
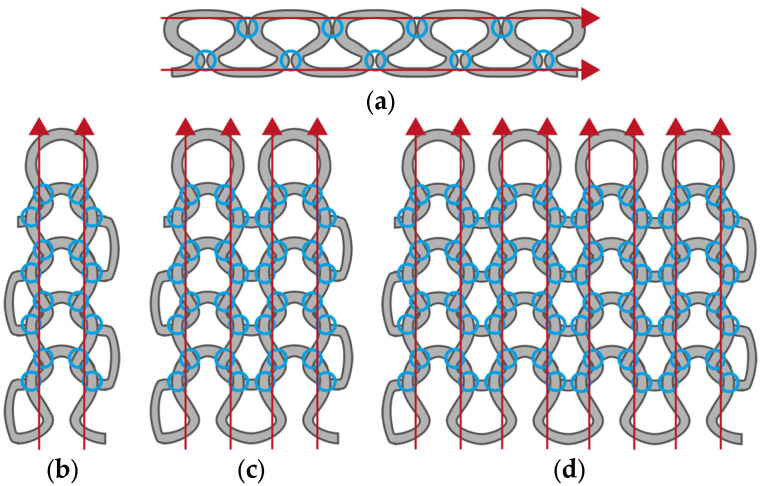
Possible contact points (blue circles) between loops and assumed current paths (red lines) for design A (**a**), design B (**b**), design C (**c**), and design D (**d**).

**Figure 9 micromachines-15-00222-f009:**
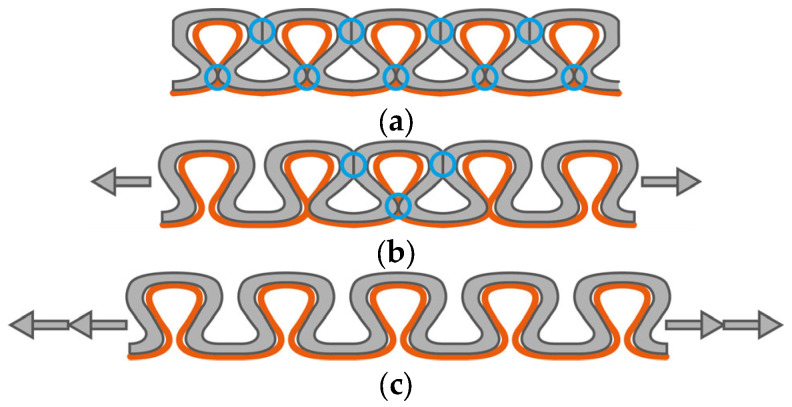
Conductive yarn (grey) plated with elastic yarn (orange): possible contact points (blue circles) between loops for course-wise integration in unloaded state (**a**), semi-loaded state (**b**), and loaded state (**c**).

**Figure 10 micromachines-15-00222-f010:**
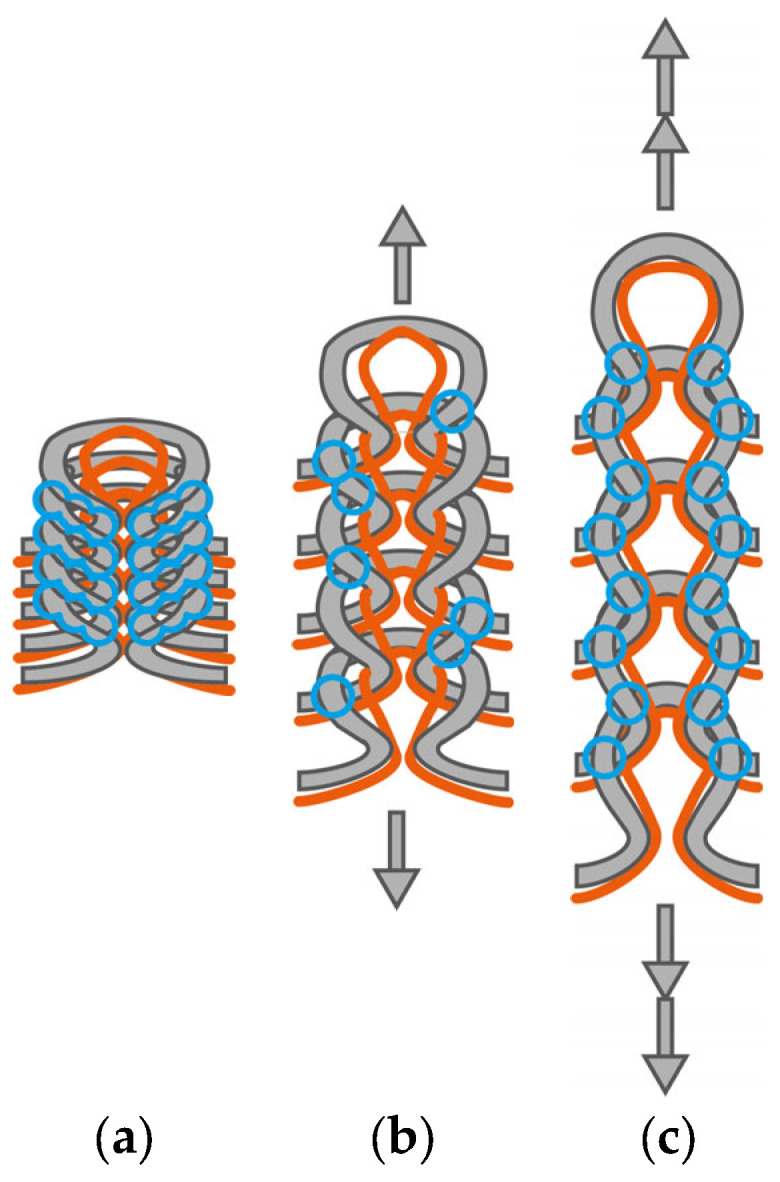
Conductive yarn (grey) plated with elastic yarn (orange): possible contact points (blue circles) between loops for wale-wise integration in unloaded state (**a**), semi-loaded state (**b**), and loaded state (**c**).

**Table 1 micromachines-15-00222-t001:** Recommended sensor design and additional benefits.

	Knitting Structure	Sensor Yarn	Sensor Geometry
Sensor design	plain right–left weft-knitted structure (single jersey)	silver-plated yarn in combination with elastic yarn	linear-shaped rectangle with a narrow sensor width
Additional benefit for the later leggings	Greater design freedom in the manufacturing process due to the use of just one needle bed	comfortable and close fit of the garment due to the use of elastic yarn	unobtrusive appearance due to the use of narrow sensor width

**Table 2 micromachines-15-00222-t002:** Overview of materials and bindings of basic weft-knitted structure and textile sensors.

Materials	Sensor Binding				
Sensor yarn: Silver-tech+ 150 (red area) Base yarn: TENCEL™ lyocell fiber yarn (yellow area) Single covered elastic yarn: Lycra/PA66 (red and yellow area)		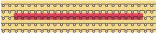 Design A			Integration direction course-wise A: 100 wales × 1 courses
		 Design B	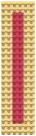 Design C	 Design D	Integration direction wale-wise B: 1 wale × 200 courses C: 2 wales × 200 courses D: 4 wales × 200 courses

**Table 3 micromachines-15-00222-t003:** Determined elongation of a weft-knitted structure fixed over knee joint during leg angulation.

Leg Position, Bent Angle	Length of Knit Structure L_i_ [cm]		Strain of Knit Structure ε_m_ = (L_i_ − L_1_)/L_1_ [%]
Unbent (0°)	L_1_	12.0	
Bent by 45°	L_2_	15.1	26
Bent by 90°	L_3_	16.3	36
Fully bent (130°)	L_4_	18.0	50

**Table 4 micromachines-15-00222-t004:** Average and standard deviation of base resistance R_0_ for different designs of weft-knitted strain sensors.

R_0_ [Ω/cm]	Design A	Design B	Design C	Design D
Average R_0_	14.69	3.77	2.51	1.61
SD	0.38	0.14	0.11	0.04

**Table 5 micromachines-15-00222-t005:** Gauge factor values and standard deviation (SD) of the different sensor designs.

Gauge Factor GF	Design A	Design B	Design C	Design D
Average	0.04	1.65	2.41	2.48
SD	0.11	0.06	0.11	0.13

## Data Availability

All data are available from the authors upon request.

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
