# Peer review of "Novel Weft-Knitted Strain Sensors for Motion Capture [Author-notes fn1-micromachines-15-00222]"

_micromachines, 2024, doi:10.3390/mi15020222_

Round 1
Reviewer 1 Report
Comments and Suggestions for Authors
This version of the paper is much better written and the the concept is explained quite clearly. I only recommend minor changes as follow :
- Figures 3 to 6 are of poor quality. They need to be redrawn and combined. The text on the figures needs to be easily readable
- Add references supporting the following assumptions : 130 degrees of maximum angle on knees // corresponding strain of 50%
- Aging of the device needs to be tested (or at least discussed) with regard to Temperature and relative humidity changes, as well as repeatability of the strains cycles
- How does the sensor integration and area influence the mechanical properties of fabric ? Determining true-stress/true strain curves for each system is advised
-The conductive coatings of the different fibers need to be analyzed by SEM and or EDX in terms of homogeneity and thickness before and after utilization of the strain sensor. Such results could explain the sensors' performance
Reviewer 2 Report
Comments and Suggestions for Authors
The authors are requested to respond to the below comments point by point.
Some graphical figures can be included to show the full flexion of the knee joint and the position of the weft-knitted strain sensors on the lower legs.
What is the cost involved in developing a single piece of Weft-knitted strain sensor? It is cheaper than using some flexible polymer materials like (PDMS, PET, etc.).
Please specify the non-linearity and hysteresis values for designs C and D. What can be done to reduce these characteristics?
Why the sensor in design A is integrated in a course-wise direction and has a sensor width of only one
loop? The authors would have tried some more designs with sensor widths of two and four loops.
Are any temperature studies performed on the Weft-knitted strain sensor to identify the variation of the resistance change due to body or ambient temperature?
The absorption of body sweat by the fabric will affect the base resistance of the strain sensor in a practical case? Please explain
Reviewer 3 Report
Comments and Suggestions for Authors
The paper presented an alternative approach of weft-knitted strain sensors that are directly integrated in the knee area of a functional legging suitable for daily use. My additional comments are:
1. It is necessary to summarize the originality and novelty of this work. I will suggest the authors to re-organized the introduction part to more focus on it.
2. As a wearable device, how about the practically, such as the washability and durability, of the sensor?
3. Some related references could be mentioned, such as ACS Appl. Mater. Interfaces 13 (2021) 17110-17117, Adv. Fiber Mater. 4 (2022) 1005–1026.
Comments on the Quality of English LanguageThe English Language should be furthrt improved and checked carefully, such as "For the gait analyses suitable conductive yarn-based resistive strain sensors on the knee area are needed.".
Round 2
Reviewer 1 Report
Comments and Suggestions for Authors
Authors have addressed most of my concerns